# Predictive Model for Preeclampsia Combining sFlt-1, PlGF, NT-proBNP, and Uric Acid as Biomarkers

**DOI:** 10.3390/jcm12020431

**Published:** 2023-01-05

**Authors:** Carmen Garrido-Giménez, Mónica Cruz-Lemini, Francisco V. Álvarez, Madalina Nicoleta Nan, Francisco Carretero, Antonio Fernández-Oliva, Josefina Mora, Olga Sánchez-García, Álvaro García-Osuna, Jaume Alijotas-Reig, Elisa Llurba

**Affiliations:** 1Department of Obstetrics and Gynecology, Maternal-Fetal Medicine Unit (Hospital de la Santa Creu i Sant Pau, Sant Antoni Maria Claret, 167), Universitat Autònoma de Barcelona, 08025 Barcelona, Spain; 2Women and Perinatal Health Research Group, Institut d’Investigació Biomèdica Sant Pau (IIB SANT PAU), Sant Quintí 77–79, 08041 Barcelona, Spain; 3Primary Care Interventions to Prevent Maternal and Child Chronic Diseases of Perinatal and Developmental Network (SAMID-RICORS, RD21/0012) and Maternal and Child Health Development Network (SAMID, RD16/0022), Instituto de Salud Carlos III, 28040 Madrid, Spain; 4Clinical Biochemistry, Laboratory Medicine, Hospital Universitario Central de Asturias and Department of Biochemistry and Molecular Biology, Universidad de Oviedo, 33011 Oviedo, Spain; 5Clinical Biochemistry, Hospital de la Santa Creu i Sant Pau, Universitat Autònoma de Barcelona, 08025 Barcelona, Spain; 6Cátedra de Inteligencia Analítica, Universidad de Oviedo, 33011 Oviedo, Spain; 7Systemic Autoimmune Disease Unit, Internal Medicine Department, Vall d’Hebron University Hospital, Departament de Medicina de la Universitat Autònoma de Barcelona, 08025 Barcelona, Spain; 8Systemic Autoimmune Diseases Research Group, Vall d’Hebron Research Institute/Vall d’Hebron Hospital, 08025 Barcelona, Spain

**Keywords:** angiogenic factors, machine-learning, N-terminal pro-brain natriuretic peptide (NT-proBNP), placental growth factor (PlGF), prediction, preeclampsia, soluble fms-like tyrosine kinase 1 (sFlt-1), uric acid

## Abstract

N-terminal pro-brain natriuretic peptide (NT-proBNP) and uric acid are elevated in pregnancies with preeclampsia (PE). Short-term prediction of PE using angiogenic factors has many false-positive results. Our objective was to validate a machine-learning model (MLM) to predict PE in patients with clinical suspicion, and evaluate if the model performed better than the sFlt-1/PlGF ratio alone. A multicentric cohort study of pregnancies with suspected PE between 24^+0^ and 36^+6^ weeks was used. The MLM included six predictors: gestational age, chronic hypertension, sFlt-1, PlGF, NT-proBNP, and uric acid. A total of 936 serum samples from 597 women were included. The PPV of the MLM for PE following 6 weeks was 83.1% (95% CI 78.5–88.2) compared to 72.8% (95% CI 67.4–78.4) for the sFlt-1/PlGF ratio. The specificity of the model was better; 94.9% vs. 91%, respectively. The AUC was significantly improved compared to the ratio alone [0.941 (95% CI 0.926–0.956) vs. 0.901 (95% CI 0.880–0.921), *p* < 0.05]. For prediction of preterm PE within 1 week, the AUC of the MLM was 0.954 (95% CI 0.937–0.968); significantly greater than the ratio alone [0.914 (95% CI 0.890–0.934), *p* < 0.01]. To conclude, an MLM combining the sFlt-1/PlGF ratio, NT-proBNP, and uric acid performs better to predict preterm PE compared to the sFlt-1/PlGF ratio alone, potentially increasing clinical precision.

## 1. Introduction

Preeclampsia (PE) is a pregnancy-related hypertensive and multisystemic disorder that affects 2–5% of pregnancies worldwide [1]. Although obstetrical care has significantly improved and reduced its mortality, it remains a leading cause of maternal morbidity and pregnancy complications such as preterm delivery, intrauterine growth restriction (IUGR), placental abruption, stillbirth, and perinatal morbidity/mortality due to prematurity [2]. Although the pathophysiology of PE is not fully understood, it is well known that it is a placental disorder with impaired trophoblast invasion and differentiation [3] that leads to an unbalance of angiogenic and antiangiogenic factors [4]. Soluble fms-like tyrosine kinase-1 (sFlt-1, an inhibitor of vascular endothelial growth factor) is responsible for maternal dysfunction, causing peripheral vasoconstriction in an attempt to raise maternal blood pressure [5]. Elevated levels of sFlt-1, reduced levels of placental growth factor (PlGF), and an increased sFlt-1/PlGF ratio have been reported both in women with established PE, and before clinical development of the disease [6,7]. A high sFlt-1/PlGF ratio seems to be a better predictor of disease severity than either marker alone [8,9,10], and these findings have led to incorporation of the sFlt-1/PlGF ratio into clinical practice to improve diagnosis and prognosis of PE [11]. It is widely accepted that the cut-off value of <38 for the sFlt-1/PlGF ratio between 24^+0^ and 36^+6^ weeks of gestation rules out PE in patients with clinical suspicion for up to four weeks [12], and the use of this ratio is cost effective [13]. However, evidence is more limited regarding management and prognosis of women that present an abnormally high sFlt-1/PlGF ratio during pregnancy.

Pregnancy promotes several changes that are stressful to the cardiovascular system in order to maintain utero–placental circulation: maternal cardiac output and heart rate increase, while blood pressure and vascular resistances decrease [14]. Cardiovascular adaptation in pregnancy is abnormal in patients with PE due to cardiac diastolic dysfunction [15]. As cardiovascular changes represent a stressful scenario for the heart, cardiac myocytes respond, producing cardiac damage markers. In particular, N-terminal pro-brain natriuretic peptide (NT-proBNP), which is used as a biomarker for heart failure in non-obstetric populations [16], has been described with higher concentrations in preeclamptic pregnancies in response to abnormal cardiovascular adaptation. Furthermore, there is a correlation between NT-proBNP levels and severity of PE [17], and higher levels are described in early vs. late-onset PE [18]. Thus, NT-proBNP could be a useful tool for the evaluation of PE and prediction of maternal cardiovascular complications reflecting cardiac changes. Serum urid acid is consistently elevated in PE secondary to reduced glomerular filtration, increased resorption, and decreased secretion in the proximal tubule, although the reasons for such an elevation are incompletely understood [19]. Thus, hyperuricemia has been classically considered a good biomarker for PE, since high concentrations have been associated with more severe disease and adverse outcomes at time of delivery [20].

Neither NT-proBNP nor uric acid individually have been shown to be good predictors for PE [21], but models combining these biomarkers show promise. Lafuente-Ganuza et al. [22], published in 2020 a predictive machine-learning algorithm for early-onset PE using a combination of the sFlt-1/PlGF ratio, NT-proBNP and uric acid as biomarkers, with apparent better positive predictive values (PPV) than the sFlt-1/PlGF ratio alone. External validation is necessary to determine reproducibility for a prediction model and applicability to different populations. For this study, our primary objectives were, first, to perform external validation of this machine-learning model (MLM) to predict preterm PE in patients with clinical suspicion and, second, to study if the model performed better than the sFlt-1/PlGF ratio alone.

## 2. Materials and Methods

### 2.1. Study Design

Our real-world dataset included pregnant women with suspected PE between 24^+0^ and 36^+6^ weeks admitted to the Obstetrics Department of seven Spanish University Hospitals between March 2018 and December 2020. Patients were part of the EuroPE study cohort, a randomized open-label controlled trial to evaluate if the incorporation of sFlt1/PlGF ratio in diagnosis and classification of PE improved maternal and perinatal outcomes in women with suspicion of the disease (NCT03231657). Blood samples were obtained at inclusion (upon suspicion of PE), and multiple samples per patient were allowed but restricted to one sample per gestational week. The study protocol was approved by the institutional Ethics Committee (IIBSP-EUR-2017-20) and all patients provided written informed consent. Exclusion criteria were pregnant women outside 24^+0^ and 36^+6^ weeks, multiple pregnancies, fetal chromosomal or congenital anomalies, those lost to follow-up and conditions that required immediate delivery (eclampsia, pulmonary edema, uncontrolled hypertension, severe visual disturbances, severe headache, fetal demise, and non-reassuring fetal status).

### 2.2. Diagnostic Criteria

Criteria for diagnosis of PE were those of the International Society for the Study of Hypertension in Pregnancy [23]. PE was defined as a previously normotensive woman who presented with systolic or diastolic blood pressure > 140/90 mmHg, measured twice (at least 4 h apart), and proteinuria > 300 mg in a 24-h urine specimen or 2+ protein on dipsticks in urine after 20 weeks’ gestation. A diagnosis of PE before 33^+6^ weeks of gestation was considered early-preterm PE, and between 34^+0^ and 36^+6^ weeks, late-preterm PE.

Suspected PE was considered upon high blood pressure or aggravation of pre-existing hypertension, new onset of proteinuria or aggravation of pre-existing proteinuria, or one or more preeclampsia-related symptoms such as epigastric pain, severe edema (face, hands, feet), persistent headache, visual disturbances, sudden weight gain (>1 kg/week in the third trimester). It was also suspected when low platelets (<100.000) or elevated liver enzymes were detected in blood analysis, as well as when abnormal uterine perfusion was detected by Doppler sonography with mean pulsatility index > 95th percentile in the second trimester and/or bilateral uterine artery notching. PE was also suspected when IUGR was detected, and was defined as an estimated fetal weight below the 3rd centile for gestational age (GA) according to local reference curves [24], or an estimated fetal weight below the 10th centile together with umbilical artery or mean uterine arteries pulsatility index above the 95th centile [25].

Adverse maternal outcome was defined as admission to the intensive care unit (ICU), eclampsia, placental abruption, disseminated intravascular coagulation, pulmonary edema, and HELLP syndrome. HELLP syndrome is a severe form of PE diagnosed when hemolysis, elevated liver enzymes (>100 U/L) and low platelet counts < 100 × 10^9^ /L were detected, with or without proteinuria or severe hypertension [26]. Adverse perinatal outcomes were defined as preterm delivery before 34^+0^ weeks, IUGR, stillbirth, and placental abruption. Adverse neonatal outcomes included neonatal ICU admission for >48 h, proven and/or suspected sepsis, respiratory distress syndrome, intraventricular hemorrhage (grades II–VI), necrotizing enterocolitis, retinopathy, and bronchopulmonary dysplasia.

### 2.3. Laboratory Methods

Blood samples were collected in serum separator tubes and centrifuged at 3000× *g* for 15 min. Serum concentrations of PlGF, sFlt-1, and NT-proBNP were measured using automated electrochemiluminescence immunoassays on the Roche Cobas^®^ e601 platform (Roche Diagnostics GmbH, Mannheim, Germany) with a turnaround time of 18 min for PlGF and sFlt-1, and a turnaround time of 9 min for NT-proBNP. Serum concentration of uric acid was measured using an automated colorimetric uricase method on the Abbott Alinity^®^ c platform (Abbott Laboratories, Chicago, IL, USA) with a turnaround time of 10 min. Product codes of reagents are 08P5620 for uric acid, 09315284190 for NT-proBNP, 07027818190 for sFlt-1, and 07027648190 for PlGF. The measuring ranges were 3–10,000 pg/mL for PlGF, 10–85,000 pg/mL for sFlt-1, 10–35,000 ng/L for NT-proBNP, and 60–1950 μmol/L for uric acid. The limits of quantification were 10 pg/mL for PlGF, 15 pg/mL for sFlt-1, 50 ng/L for NT-proBNP, and 10 μmol/L for uric acid. No high-dose hook effect has been described for concentrations up to 100,000 pg/mL for PlGF, 200,000 pg/mL for sFlt-1, 300,000 ng/L for NT-proBNP. Intra- and interassay coefficients of variation, evaluated with PreciControl Multimarkers 1 and 2 (Roche Diagnostics) for PlGF and sFlt-1, with PreciControl Cardiac 1 and 2 (Roche Diagnostics) for NT-proBNP, and with Multichem S Plus 1, 2 and 3 (Technopath) for uric acid, were found to be <5% in all assays.

### 2.4. Statistical Analysis

Demographic data were analyzed using the IBM SPSS Statistics 26 statistical package. Variables studied were tested for normal distribution using the Kolmogorov–Smirnov test. Comparisons between study groups were performed with analysis of variance (ANOVA) or Chi-squared test when appropriate, and are presented as mean (standard deviation) or percentage (*n*) *p*-values below 0.05 were considered statistically significant for all tests performed.

The MLM to predict PE included six predictors, as previously published [22]. Briefly, GA at admission, chronic hypertension, and biomarker serum levels (sFlt-1, PlGF, NT-proBNP, uric acid), corrected for GA at sampling, were included. Pregnancy data and outcomes were blinded to the professional who applied the random forest-based supervised MLM to predict the risk or probability of PE (low, moderately low, moderately high, or high) of a patient with clinical suspicion. We defined a patient as negative (no PE) if none of the patient’s repeated measurements presented a moderately high risk or above. Otherwise, the patient was considered positive, and with high probability of developing PE. True positive and true negative patients are those coincident with the final decision of the clinician. The predictive model and the decision thresholds for being positive (high or very high) or being negative (low or very low) are found in the Appendix A. Scikit-learn (version 0.23.2), an open-source python library, was used to support the best machine-learning practices for setting up the predictive model of the software. The *p*-values and 95% confidence intervals were calculated by using the bootstrap method. Statistical significance of sensitivity, specificity, predictive values, and area under curve (AUC) were calculated using the 95% percentile bootstrap confidence intervals with 10,000 bootstrap samples.

## 3. Results

### 3.1. Characteristics of the Study Population

A total of 792 women with suspected PE were initially recruited. A total of 4 twin pregnancies, 9 pregnancies lost to follow-up, and 182 pregnancies included at term were excluded, so the final analysis included 597 participants and 936 serum samples, obtained between 24^+0^ and 36^+6^ weeks. The global incidence of PE was 34.7% (207/597): 90 women (15.1%) developed early-preterm PE (<34^+0^), 67 (11.2%) had late-preterm PE (34–36^+6^), 50 (6.3%) had term PE (≥37^+0^), and 390 women (65.3%) did not develop PE.

Table 1 shows the demographic, clinical, and perinatal characteristics of the study population, divided into those women that developed early-preterm PE, late-preterm PE, and those without PE. There were no differences in baseline characteristics between these groups. Women with PE had significantly higher levels of sFlt-1, sFlt-1/PlGF ratio, NT-proBNP, and uric acid compared to those without PE (*p* < 0.001 for all variables mentioned), and patients with early-preterm PE had higher levels than those with late-preterm PE. GA at delivery was significantly earlier in the early-preterm PE group. The mode of delivery showed a significantly higher prevalence of caesarean section and maternal admission to the obstetric ICU in the early-preterm PE group. Regarding neonatal outcomes, birth weight was significantly lower in the early-preterm PE group, with higher rate of IUGR, as well as lower Apgar test, and higher rate of admission to the neonatal ICU and of adverse neonatal outcomes.

### 3.2. Predictive Model Results

Table 2 and Figure 1 show validation of the MLM compared to the sFlt-1/PlGF ratio alone in predicting preterm PE. A decreased false-positive rate was observed with this model compared to the sFlt-1/PlGF ratio alone (23% vs. 41%, respectively). There was also decreased false-negative rate with the model compared to the sFlt-1/PlGF ratio alone (29% vs. 32%, respectively), and to those patients with a clinical diagnosis of PE, but with normal values of biomarkers. Table 3 shows the validation models for predicting early-preterm and late-preterm PE individually. The complete ROC curve analysis comparing the model to the sFlt-1/PlGF ratio is shown in Appendix A.

The ROC curves of the combined model and the sFlt-1/PlGF ratio for predicting preterm PE within 1 and 3 weeks of clinical suspicion are observed in Figure 2. Results of specificity, sensitivity, and predictive values comparing both tests for predicting early-preterm PE and late-preterm PE within 1 and 2 or 3 weeks of clinical suspicion are shown in Table 4. ROC curves show that the algorithm performed better as the delivery date grew closer, within one and three weeks, and are shown as Appendix A.

## 4. Discussion

### 4.1. Principal Findings

Our study demonstrates the high performance of a predictive model combining the sFlt-1/PlGF ratio, NT-proBNP, and uric acid to rule in and rule out preterm PE in women with clinical suspicion. The PPV (to develop PE in the subsequent 6 weeks) and specificity (false positives) of the model were significantly better than the sFlt-1/PlGF ratio alone. Although differences were not significant, sensitivity and NPV of the model to rule out PE were slightly better than the sFlt-1/PlGF ratio alone. These differences were observed mainly in early-preterm PE, but also in late-preterm PE, and the model performed better as the delivery date approached. The short-term prediction of early-onset and late-preterm PE within 1 week was also better with the combined model.

### 4.2. Interpretation of Results and Comparison with Existing Literature

The sFlt-1/PlGF ratio is a good predictor of PE and serious pregnancy complications, since it increases before the first clinical symptoms appear [4]. Other biomarkers used individually have not shown to improve the predictive results of the sFlt-1/PlGF ratio [9]. We and others have previously demonstrated that the sFlt-1/PlGF ratio is increased in early and severe cases of placental insufficiency (i.e., IUGR with or without PE) because of increased levels of sFlt-1 [4,10]. It is known that high values of the sFlt-1/PlGF ratio are associated with shorter intervals to delivery [8], but hospitalization and intense monitoring in test-positive patients are often required because of high false positives. The reason why an antiangiogenic state does not always result in development of PE is still unclear, but maternal predisposition is probably necessary, added to a severe and prolonged endothelial insult.

NT-proBNP has shown higher concentrations in women with hypertensive disorders of pregnancy [18,27], especially those complicated with severe and preterm PE [27,28]. A recent study reported mean NT-proBNP levels of 349.1 ± 93.5 pg/mL in PE without severe clinical signs and 725.3 ± 290.5 pg/mL in severe PE [29]. In our data, NT-proBNP levels were also higher in preterm PE, with early-preterm showing the highest values, and higher levels were observed in the most severe cases. NT-proBNP levels are higher in pregnant women with chronic hypertension [30,31]; therefore, this was taken into account in the MLM of our study and the necessary adjustments were performed.

The prospective, multicenter, and observational PROGNOSIS study [9] reported that an sFlt-1/PlGF ratio of 38 was the optimal cut off in pregnancies between 24^+0^ and 36^+6^ weeks to rule out PE in women with clinical suspicion. This cut off performed with an NPV of 99.3% (95% CI, 97.9 to 99.9) within 1 week, and remained high at 2 (97.9%), 3 (95.7%), and 4 weeks (94.3%) after testing [12]; that is, a woman with results in this range was extremely unlikely to progress to PE or HELLP within the next month. In our study, the combined model rules out preterm PE in the next 6 weeks with 93.7% probability, slightly better than the sFlt-1/PlGF ratio alone, and performs better for early-preterm PE, ruling out with a 96% probability. Although the NPV of the sFlt-1/PlGF ratio to rule out early-preterm PE within 1 week was slightly lower than that reported in the PROGNOSIS study, this improved with the MLM (96.5% to 97.6%).

On the other hand, an sFlt-1/PlGF ratio > 38 was characterized by poor PPV: within 1 week 16.7% (95% CI, 12.3–21.9), and in the next 4 weeks 36.7% (95% CI, 28.4 to 45.7%), with 66.2% sensitivity (95% CI, 54.0–77.0) and 83.1% specificity (95% CI, 79.4–86.3) [9]. In addition, the reported ability of NT-proBNP to predict PE alone is modest, with an AUC of 0.55 [32] and 0.69 [31] in the first and third trimesters, respectively. The strategy of adding NT-proBNP to the sFlt/PlGF ratio to try to improve short-term prediction of PE has previously been reported. Lafuente-Ganuza et al. [22] identified that when maternal serum NT-proBNP value > 174 pg/mL was combined with an sFlt-1/PlGF ratio > 45, the PPV for diagnosis of early-onset PE (24^+0^ to 33^+6^weeks) was 86% (95% CI: 79.2–92.6) at any point in time, with a sensitivity of 72.5% (95% CI 70.5–81.8) and specificity of 97.7% (95% CI 96.7–98.5). Similar results have been obtained in our study. In our cohort the PPV of the model is somewhat lower to predict preterm PE in the subsequent 6 weeks, but is statistically superior to those obtained with the sFlt1/PlGF ratio alone [72.5% (95% CI 66.1–79.4) vs. 80.4% (95% CI 74.2–86.4)], with statistically better specificity. This is the first time this model was applied to pregnancies between 34^+0^ and 36^+6^ weeks but it performed better for early-preterm PE, and also surpassed the PPV and specificity of the sFlt-1/PlGF ratio alone. Our data did not allow us to perform analysis for a 4-week period, but the model’s precision for PE was also better than the sFlt1/PlGF ratio alone from 1 to 3 weeks, improving clinical decisions regarding monitoring intervals. Sabrià et al. [33] described a prediction model where the addition of NT-proBNP assessment to an sFlt/PlGF ratio ≥ 38 yielded a superior ability for detecting delivery in the subsequent week in women with suspected PE. The AUC was 0.845 (95% CI 0.7870.896), which was significantly greater than the AUC of the sFlt-1/PlGF ratio alone 0.786 (95% CI: 0.722–0.844). This means that when adding NT-proBNP to the sFlt1/PlGF ratio, a false-positive rate reduction of 18.2% could be achieved. Finally, in line with previous studies, serum levels of uric acid were higher in pregnancies complicated with PE [34,35]. Hyperuricemia precedes the onset of proteinuria and hypertension in PE, but its prognostic value is debated [36]. It seems that among women with diagnosed PE, higher levels may help to identify those who will develop a more severe disease, but a cut-off value has not been stablished.

Multiple sFlt-1/PlGF ratio cut offs have been studied to enhance the diagnostic accuracy of this maternal syndrome. The MLM produced a cut-off value of 45 for the sFlt/PlGF ratio, 174 pg/mL for NT-proBNP, and 5.6 mg/dL for uric acid for PE prediction of early-preterm PE. A retrospective cohort study [22] described that the PPV to rule in early-onset PE of the sFlt-1/PlGF ratio > 45 was 49.5% (95% CI 41.9–51.8), with a specificity of 79.8% (95% CI 78.5–83.6). A multicenter case–control study including a total of 1149 patients concluded that the sFlt-1/PlGF ratio ≥ 85 for early-onset PE and ≥110 for late-onset PE resulted in a sensitivity/specificity of 88%/99.5% and 58.2%/95.5%, respectively [37]. A recent study has reported that uric acid has similar specificity to the sFlt-1/PlGF ratio for the diagnosis of PE, although the sensitivity appears to be much lower [38]. The NT-proBNP cut-off point to associate pregnancy complications is not known. Serum NT-proBNP < 125 pg/mL excludes heart failure in non-obstetric populations [39], and it seems that maternal serum NT-proBNP levels < 40.6 pg/mL could rule out PE with a high NPV of 92% [40]. Alvarez et al. [18] described that a cut-off point of NT-proBNP 219 ng/L predicted development of an adverse outcome in pregnant women < 34 weeks, with a sensitivity of 76% and specificity of 94%. Another study of this group demonstrated similar performance in the prediction of adverse outcomes with cut-off points of 178 and 219 for the sFlt-1/PlGF ratio and NT-proBNP, with a sensitivity/specificity of 95/84% and 94/76%, respectively [18]. Therefore, it seems that maternal serum NT-proBNP levels > 129.5 g/mL would warrant close follow-up during pregnancy [27,41,42].

Pregnancies with placental-related disorders, such as isolated IUGR or placental abruption, also show increased sFlt-1/PlGF ratio levels, exceeding the cut-off points for PE diagnosis explained above, but it is unknown how these entities affect NT-proBNP levels [43]. Elevated NT-proBNP levels may reflect ventricular stress and subclinical cardiac dysfunction, worsening if IUGR is present. When PE is associated with IUGR, patients present higher peripheral vascular resistance and lower cardiac output compared to isolated IUGR [44]. Few studies have evaluated NT-proBNP levels in pregnant women with IUGR, and higher maternal serum NT-proBNP levels have been detected in pregnancies with early-onset PE, with or without IUGR, than in pregnancies with isolated IUGR [22,41]. Therefore, when a high sFlt/PlGF ratio is detected, lower levels of NT-proBNP could discriminate between those pregnant women that are not going to develop PE.

### 4.3. Clinical Implications and Future Research Directions

Clinical presentation of PE is heterogeneous, and a daily challenge to the practicing obstetrician. This diagnostic model using a combination of cardiac, renal, and placental biomarkers has been validated to predict preterm PE in patients with clinical suspicion, with better PPV, sensitivity, and specificity than the sFlt-1/PlGF ratio alone. This information could help clinicians decide which women may be followed up on safely on an outpatient basis, and which women need careful and close surveillance, and hospitalization. If expectant management is considered, this MLM could also provide more accurate and valuable information for the frequency of patient assessment and follow-up. Furthermore, a specific software could be developed and studied to calculate the risk of maternal and fetal-adverse outcomes, and indicate imminent delivery, evaluating also the cost effectiveness of the routinely incorporation of uric acid and NT-proBNP in women with suspected PE.

### 4.4. Strengths and Limitations

This study externally validates an MLM previously published to predict PE. This is the first study that improves the sFlt-1/PlGF ratio’s prognosis of PE using other related maternal serum PE biomarkers. It is a nested study, with a very strict selection process for patients, prospectively included in the database but not managed based on these results; therefore, it has no bias with regard to our findings. However, more data from randomized trials is needed to establish whether the use of this algorithm in clinical practice could really reduce unnecessary hospitalization and costs. Another limitation of this study is that the different cut-off values proposed for the sFlt-1/PlGF ratio can only be applied when using Elecsys immunoassays, since differences have been reported with other brands or platforms.

## 5. Conclusions

An MLM combining the sFlt-1/PlGF ratio, NT-proBNP, and uric acid performs better to predict preterm PE compared to the sFlt-1/PlGF ratio alone, potentially increasing clinical precision, decreasing false-positive rates, and increasing PPV and specificity. This model also performed better than the sFlt-1/PlGF ratio for prediction of PE within 1 and 3 weeks. These results could avoid unnecessary interventions in women with suspected PE. Our work highlights the need of further studies combining different biomarkers and supports the use of NT-proBNP and uric acid added to maternal serum angiogenic factors to increase PE prediction.

## Figures and Tables

**Figure 1 jcm-12-00431-f001:**
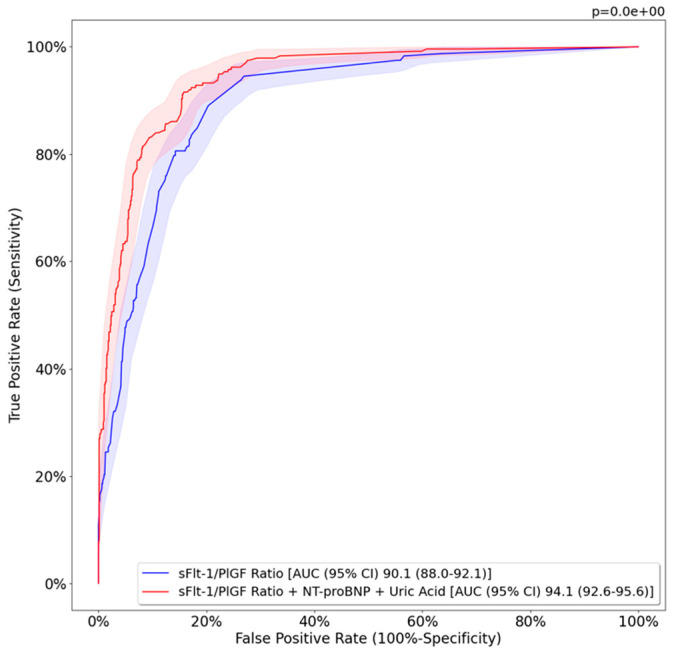
Performance of studied models for predicting preterm PE. sFlt-1, soluble fms-like tyrosine kinase 1; PlGF, placental growth factor; NT-proBNP, N-terminal pro-brain natriuretic peptide; AUC, area under the curve; CI, confidence interval. Preterm PE was defined as PE below 37^+0^ weeks.

**Figure 2 jcm-12-00431-f002:**
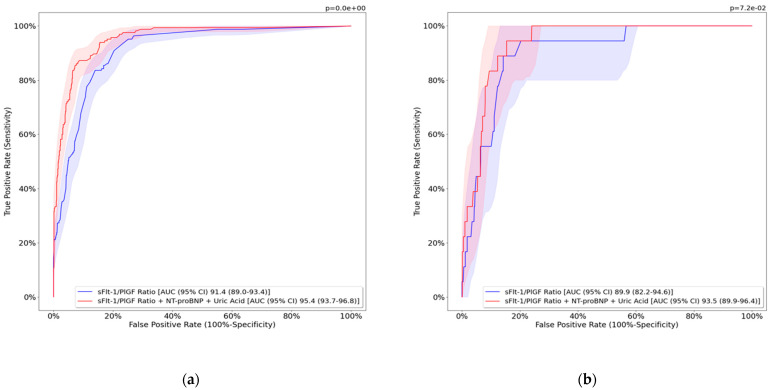
Performance of studied models for predicting preterm PE within 1 and 3 weeks to delivery. (**a**) Prediction of preterm PE within 1 week to delivery, (**b**) prediction of preterm PE within 3 weeks to delivery. sFlt-1, soluble fms-like tyrosine kinase 1; PlGF, placental growth factor; NT-proBNP, N-terminal pro-brain natriuretic peptide; AUC, area under the curve; CI, confidence interval. Preterm PE was defined as PE below 37^+0^ weeks.

**Table 1 jcm-12-00431-t001:** Demographic, clinical, and perinatal characteristics of the study population.

	No Preeclampsia (*n* = 390)	Early-Preterm Preeclampsia (*n* = 90)	Late-Preterm Preeclampsia (*n* = 67)	*p*-Value
**Maternal characteristics**				
Age (years)	34.02 ± 6	35.5 ± 5.9	34.8 ± 7.6	0.422
Ethnicity Caucasian Black Asian	88.1 (258) 7.2 (21) 4.8 (14)	76.5 (26) 11.8 (4) 11.8 (4)	94.2 (49) 1.9 (1) 3.8 (2)	0.143
Smoking	7.6 (22)	8.8 (3)	0 (0)	0.112
Nulliparity	39.8 (117)	55.9 (19)	44.2 (23)	0.184
**Maternal morbidity**				
Diabetes mellitus type 1	1.4 (4)	2.9 (1)	3.8 (2)	0.415
Diabetes mellitus type 2	1.7 (5)	0 (0)	5.8 (5)	0.114
Hypertension	17 (50)	17.6 (6)	13.5 (7)	0.806
Cardiovascular disease	0.7 (2)	0 (0)	0 (0)	0.745
Renal disease	2.4 (7)	2.9 (1)	1.9 (1)	0.955
**Biomarker data**				
GA at sampling (weeks)	33.6 ± 3.3	30.8 ± 2.1 *	34.9 ± 1.2 ^¥^	<0.001
sFlt-1 (pg/mL)	3252 ± 2815.2	15043.1 ± 9289 *	12044.8 ± 9167.7 *^,¥^	<0.001
PlGF (pg/mL)	361.2 ± 394.7	66.9 ± 84.4	83.4 ± 72.9	0.227
sFlt-1/PlGF ratio	28.9 ± 56.8	443.7 ± 329.1 *	220.8 ± 229.9 *^,¥^	<0.001
NT-proBNP (ng/L)	44.3 ± 46.3	883.5 ± 2391.5 *	261.6 ± 247.2 *^,¥^	<0.001
Uric acid (mg/dL)	3.9 ± 1	5.9 ± 1.4 *	5.8 ± 1.5 *^,¥^	<0.001
**Pregnancy outcomes**				
GA at delivery (weeks)	38.3 ± 2.3	31.4 ± 2.1 *	35.3 ± 1 *^,¥^	<0.001
Mode of delivery Vaginal Operative vaginal Cesarean section	53.3 (154) 6.6 (19) 39.9 (115)	11.8 (4) * 0 (0) 88.2 (30) *	32.7 (17) * 0 (0) 67.3 (35) *	<0.001
Maternal admission to OICU	0.7 (2)	70.6 (24) *	63.5 (33) *	<0.001
**Neonatal outcomes**				
Birth weight (grams)	2755.4 ± 654.5	1315.5 ± 372 *	2249.1 ± 458.1 *^,¥^	<0.001
IUGR	23.7 (72)	82.9 (29) *	43.6 (24) *^,¥^	<0.001
1′ Apgar score	8 ± 1	6 ± 2 *	8 ± 0 ^¥^	<0.001
5′ Apgar score	9 ± 0	8 ± 1 *	9 ± 1 ^¥^	<0.001
Umbilical artery pH	7.2 ± 0.6	7.3 ± 0.1	7.2 ± 0.1	0.943
Umbilical vein pH	7.3 ± 0.1	7.3 ± 0.1	7.3 ± 0.1	0.05
Admission to NICU	13 (37)	97 (32) *	52.9 (27) *^,¥^	<0.001
Adverse neonatal outcome	11.3 (32)	87.9 (29) *	28.8 (15) *^,¥^	<0.001

Data shown as mean ± SD or %(*n*). GA, gestational age; IUGR, intrauterine growth restriction; NT-proBNP, N-terminal pro-brain natriuretic peptide; NICU, neonatal intensive care unit; OICU, obstetric intensive care unit; PlGF, placental growth factor; sFlt-1, soluble fms-like tyrosine kinase 1. *p*-values obtained by ANOVA or Chi-squared test, where appropriate, and comparisons were performed between groups. * *p*-value < 0.05 compared to no preeclampsia; ^¥^
*p*-value < 0.05 compared to early-preterm PE.

**Table 2 jcm-12-00431-t002:** Validation of models for predicting preterm PE.

	sFlt-1/PlGF Ratio	sFlt-1/PlGF Ratio + NT-proBNP + Uric Acid	*p*-Value
Sensitivity (%)	77.5 (71.9–83.0)	79.6 (74.4–84.5)	0.210
Specificity (%)	91.0 (89.0–93.0)	94.9 (93.4–96.5)	<0.05
PPV (%)	72.8 (67.4–78.4)	83.1 (78.5–88.2)	<0.05
NPV (%)	92.8 (91.0–94.7)	93.7 (92.0–95.3)	0.140

sFlt-1, soluble fms-like tyrosine kinase 1; PlGF, placental growth factor; NT-proBNP, N-terminal pro-brain natriuretic peptide; PPV, positive predictive value; NPV, negative predictive value. Preterm PE was defined as PE below 37^+0^ weeks. Sensitivity to rule in was calculated as the proportion of positives that were correctly classified as patients in whom preeclampsia developed at any time between 24 and 36^+6^ weeks of gestation. Specificity to rule in was calculated as 1-proportion of positives that were incorrectly classified as patients in whom preeclampsia developed any time between 24 and 36^+6^ weeks of gestation.

**Table 3 jcm-12-00431-t003:** Validation of models for predicting early-preterm and late-preterm PE.

Biomarkers	sFlt-1/PlGF Ratio	sFlt-1/PlGF Ratio + NT-proBNP + Uric Acid
**Early-preterm PE**		
Sensitivity (%)	82.2 (76.3–88.5)	86.7 (81.8–92.6)
Specificity (%)	90.8 (88.4–93.5)	93.8 (91.7–95.9) *
PPV (%)	72.5 (66.1–79.4)	80.4 (74.2–86.4) *
NPV (%)	94.5 (92.6–96.6)	96.0 (94.3–97.8)
**Late-preterm PE**		
Sensitivity (%)	63.5 (53.1–75.0)	63.5 (53.1–75.0)
Specificity (%)	90.2 (87.6–92.8)	96.0 (94.2–97.8) *
PPV (%)	55.0 (45.7–64.9)	75.0 (65.2–86.2) *
NPV (%)	92.9 (90.7–95.5)	93.3 (91.1–95.7)

sFlt-1, soluble fms-like tyrosine kinase 1; PlGF, placental growth factor; NT-proBNP, N-terminal pro-brain natriuretic peptide; PPV, positive predictive value; NPV, negative predictive value. Early-preterm PE was defined as PE below 34^+0^ weeks, late-preterm PE was defined as PE between 34^+0^ and 36^+6^ weeks. Sensitivity to rule in was calculated as the proportion of positives that were correctly classified as patients in whom preeclampsia developed at any time between 24 and 36^+6^ weeks of gestation. Specificity to rule in was calculated as 1-proportion of positives that were incorrectly classified as patients in whom preeclampsia developed any time between 24 and 36^+6^ weeks of gestation. * *p*-value < 0.05.

**Table 4 jcm-12-00431-t004:** Prediction of early and late-preterm PE using the combined model vs. sFlt-1/PlGF ratio alone.

Early-Preterm PE	sFlt-1/PlGF Ratio	sFlt-1/PlGF Ratio + NT-proBNP + Uric Acid	Late-Preterm PE	sFlt-1/PlGF Ratio	sFlt-1/PlGF Ratio + NT-proBNP + Uric Acid
Within 1 week			Within 1 week		
Sensitivity (%)	87.0 (81.4–92.7)	90.9 (86.0–95.9)	Sensitivity (%)	65.3 (54.8–75.9)	65.3 (54.8–75.9)
PPV (%)	70.5 (63.8–77.6)	78.7 (72.3–84.9) *	PPV (%)	54.2 (44.7–63.6)	74.4 (64.3–84.0) *
Specificity (%)	90.8 (88.1–93.2)	93.8 (91.8–95.8) *	Specificity (%)	90.2 (87.6–92.9)	96.0 (94.3–97.8) *
NPV (%)	96.5 (94.9–98.2)	97.6 (96.2–98.9)	NPV (%)	93.6 (91.5–95.9)	94.0 (92.0–96.1)
Within 3 weeks			Within 2 weeks		
Sensitivity (%)	71.4 (50.0–100.0)	50 (25.0–71.4)	Sensitivity (%)	40.0 (18.2–60.0)	40.0 (18.2–60.0)
PPV (%)	26.3 (13.6–36.4)	26.9 (11.8–40.0)	PPV (%)	18.2 (5.9–27.8)	35.3 (15.4–55.6)
Specificity (%)	90.8 (88.4–93.3)	93.8 (91.7–95.9) *	Specificity (%)	90.2 (87.6–92.9) *	96.0 (94.3–97.8) *
NPV (%)	98.6 (97.7–100.0)	97.6 (96.3–99.0)	NPV (%)	96.5 (94.8–98.2)	96.7 (95.1–98.3)

sFlt-1, soluble fms-like tyrosine kinase 1; PlGF, placental growth factor; NT-proBNP, N-terminal pro-brain natriuretic peptide; PPV, positive predictive value; NPV, negative predictive value. * *p*-value < 0.05.

## Data Availability

The data presented in this study are available upon request from the corresponding author. The data are not publicly available due to privacy policies.

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
