# Peer review of "Predictive Model for Preeclampsia Combining sFlt-1, PlGF, NT-proBNP, and Uric Acid as Biomarkers"

_jcm, 2023, doi:10.3390/jcm12020431_

Round 1

Reviewer 1 Report

In this study authors evaluated the performance of a machine-learning model (MLM including sFlt-1/PlGF, NT-proBNP and uric acid) to predict PE in patients with clinical suspicion. Furthermore they investigated if the MLM-model performed better than sFlt-1/PlGF ratio alone.

The authors present an interesting and methodologically good study with an adequate number of cases. The manuscript is well written and the discussion covers the main points in considering the results. 

It would be desirable if the authors examined the predictive performance of the two models with a view to adverse perinatal and maternal outcome. Does the MLM-model also shows advantages here compared to sole use sFlt-1/PlGF? The authors should comment on this. 

Author Response

Thank you for your comment. In this study, we validated a previously published machine-learning model that was designed exclusively for diagnosis of preeclampsia. Therefore, data was analyzed in two groups regarding this outcome and women that did not develop PE have been included in the control group. Other maternal and perinatal outcomes are indeed clinically relevant, however in this validation cohort we sought to replicate the same outcomes as the initial development cohort (Lafuente-Ganuza P., Lequerica-Fernandez P., Carretero F., Escudero A.I., Martinez-Morillo E., Sabria E., Herraiz I., Galindo A., Lopez A., Martinez-Triguero M.L., et al. A more accurate prediction to rule in and rule out pre-eclampsia using the sFlt-1/PlGF ratio and NT-proBNP as biomarkers. Clin Chem Lab Med. 2020;58:399-407. doi:10.1515/cclm-2019-0939) to confirm this important contribution. Moreover, the rationale for the use of NTpro-BNP as predictor of PE is given by the pathophysiology of the disease, whereas, as far as we are aware, maternal cardiac impairment has not been observed in cases with isolated intrauterine growth restriction (Lafuente-Ganuza P., Carretero F., Lequerica-Fernandez P., Fernandez-Bernardo A., Escudero A.I., De La Hera-Galarza J.M., Garcia-Iglesias D., Alvarez-Velasco R., Alvarez F.V. NT-proBNP levels in preeclampsia, intrauterine growth restriction as well as in the prediction on an imminent delivery. Clin Chem Lab Med. 2021;59:1077-1085), preterm delivery without preeclampsia or placental abruption (Afshani N., Moustagim-Barrete A., Biccard B.M., Rodseth R.N., Dyer R.A. Utility of B-type natriuretic peptides in preeclampsia: a systematic review. Int J Obstet Anesth. 2013;22:96-103). However, we understand the Reviewer’s request to improve clinical applicability of the model. In fact, we are planning to perform further studies with a wider sample size that includes all possible placental insufficiency complications.

Reviewer 2 Report

this is an interesting study evaluating a machine-learning model (MLM) to predict PE in patients with clinical suspicion, and evaluate if the model performed better than sFlt-1/PlGF ratio alone. 

Although the manuscript is interesting it presents some flaws that must be resolved. In particular:

lines 52-59: although authors properly introduced the clinical manifestation of preeclampsia, it deserves to be pointed out that preeclampsia is also characterized by trophoblast  immaturity (see PMID: 32529396). This is an important point to highlight since it can contribute to the alterations found by the authors further highlithing their interesting results

Laboratory methods: the product codes of reagents used must be reported 

Table 1:  the GA at sampling is significantly different amoung groups. Since PlGF, Uric acid, NT-proBNP and sFlt-1 levels can change depending on the gestational age, authors must compare only samples with no GA differences at sampling otherwise the altered values found could be due to the gestational age rather than preeclampsia. 

References must follow the journal style

An accurate revision of syntax is recommended

Author Response

Comments to the Author

Although the manuscript is interesting it presents some flaws that must be resolved. In particular:

  1. lines 52-59: although authors properly introduced the clinical manifestation of preeclampsia, it deserves to be pointed out that preeclampsia is also characterized by trophoblast  immaturity (see PMID: 32529396). This is an important point to highlight since it can contribute to the alterations found by the authors further highlithing their interesting results

Thank you for this important comment; we have now included this reference in lines 27-29: “Although pathophysiology of PE is not fully understood, it is well known that it is a placental disorder with impaired trophoblast invasion and differentiation (Fantone S., Mazzucchelli R., Giannubilo SR., Ciavattini A., Marzioni D., Tossetta G. AT-rich interactive domain 1A protein expression in normal and pathological pregnancies complicated by preeclampsia. Histochem Cell Biol. 2020;154:339-346) that leads to an unbalance of angiogenic and antiangiogenic factors”.

  1. Laboratory methods: the product codes of reagents used must be reported

Thank you for your comment. We have added the information for product codes of reagents used (line 111-112: “Product codes of reagents are: 08P5620 for uric acid, 09315284190 for NT-proBNP, 07027818190 for sFlt-1 and 07027648190 for PlGF”).

  1. Table 1:  the GA at sampling is significantly different among groups. Since PlGF, Uric acid, NT-proBNP and sFlt-1 levels can change depending on the gestational age, authors must compare only samples with no GA differences at sampling otherwise the altered values found could be due to the gestational age rather than preeclampsia.

Thank you; we are fully aware that GA at sampling is significantly different among groups, however this is due to the fact that the original software was developed with a cohort with different GA between 24 and 37 weeks, and the values of angiogenic markers, NT-proBNP and uric acid have been corrected by GA at sampling in the machine-learning model. We have now detailed this so it is clearer in the Materials and Methods Section.

Lines 126-128: “The MLM to predict PE included six predictors, as previously published (Lafuente-Ganuza P., Lequerica-Fernandez P., Carretero F., Escudero A.I., Martinez-Morillo E., Sabria E., Herraiz I., Galindo A., Lopez A., Martinez-Triguero M.L., et al. A more accurate prediction to rule in and rule out pre-eclampsia using the sFlt-1/PlGF ratio and NT-proBNP as biomarkers. Clin Chem Lab Med. 2020;58:399-407. doi:10.1515/cclm-2019-0939). Briefly, GA at admission, chronic hypertension and biomarker  serum levels (sFlt-1, PlGF, NT-proBNP, uric acid), corrected for GA at sampling, were included”.

  1. References must follow the journal style.

We have verified and adapted all the references to the journal style.

  1. An accurate revision of syntax is recommended.

Thank you for your comment. Syntax has now been reviewed by an English native language speaker to fulfil the Editorial quality.

Round 2

Reviewer 2 Report

the manuscript has been significantly improved and can be accepted in the present form